# Effects of Combined Usage of Supplementary Cementitious Materials on the Thermal Properties and Microstructure of High-Performance Concrete at High Temperatures

**DOI:** 10.3390/ma13081833

**Published:** 2020-04-13

**Authors:** Dong Lu, Zhuo Tang, Liang Zhang, Jianwei Zhou, Yue Gong, Yaogang Tian, Jing Zhong

**Affiliations:** 1Institute of Intelligent Manufacturing Technology, ShenZhen Polytechnic, ShenZhen 518055, China; dongluhit@163.com (D.L.); tangzhuowhpu@163.com (Z.T.); 2School of Civil Engineering, Harbin Institute of Technology, Harbin 150000, China; 3School of Materials Science and Engineering, Chang’an University, Xi’an 710064, China; Yolandagongyue@163.com (Y.G.); xguangyao_1@163.com (Y.T.); 4School of Materials Science and Engineering, Xi’an University of Architecture and Technology, Xi’an 710055, China; jianzhucat@163.com

**Keywords:** high-performance concrete, supplementary cementitious materials, high-temperature resistance, thermal performance, thermal conductivity, microstructure

## Abstract

Concrete has low porosity and compact microstructure, and thus can be vulnerable to high temperature, and the increasing application of various types of supplementary cementitious materials (SCMs) in concrete makes its high-temperature resistant behavior more complex. In this study, we investigate the effects of four formulations with typical SCMs combinations of fly ash (FA), ultra-fine fly ash (UFFA) and metakaolin (MK), and study the effects of SCMs combinations on the thermal performance, microstructure, and the crystalline and amorphous phases evolution of concrete subjected to high temperatures. The experimental results showed that at 400 °C, with the addition of 20% FA (wt %), the thermal conductivity of the sample slightly increased to 1.5 W/(m·K). Replacing FA with UFFA can further increase the thermal conductivity to 1.7 W/(m·K). Thermal conductivity of concrete slightly increased at 400 °C and significantly reduced at 800 °C. Further, combined usage of SCMs delayed and reduced micro-cracks of concrete subjected to high temperatures. This study demonstrates the potential of combining the usage of SCMs to promote the high-temperature performance of concrete and explains the micro-mechanism of concrete containing SCMs at high temperatures.

## 1. Introduction

High-performance concrete (HPC) has been extensively used in ultra-high and ultra-large building structures owing to its high strength and excellent durability [1,2]. However, HPC is extremely vulnerable when subjected to high temperatures due to its low porosity, low permeability and compact microstructure. For example, concrete subjected to high temperature leads to cracks and reduction in residual compressive strength [3]. Another threat that a high temperature could pose to concrete is explosive spalling [4,5]. Further, for a concrete structure, tensile strength failures under high temperatures could result in catastrophic consequences since it usually happens abruptly without any omen [5]. These all hinder the application of HPC in certain areas [3,4,5].

Under high-temperature conditions, cement-based materials are not stable due to their internal physical/chemical changes and formation of cracks. The disappearance of ettringite (AFt) at 100 °C, the decomposition of Calcium Silicate Hydrate (C-S-H) gel and CH at 400–600 °C, and the transformation of C-S-H gel into the nesosilicate form at 750 °C, results in the degradation and damage of concrete [6,7,8]. Similar conclusions have been illustrated by other studies [3,9]. Further, a large number of researchers indicated that the heating temperature [3,9,10], heating rate [11] and cooling regimes [12,13] all have a significant influence on the microstructure and performance of concrete.

Supplementary cementitious materials (SCMs) contribute to improve the performance of concrete at high temperatures owing to their pozzolanic effects and super micro-filling capacity [14]. Furthermore, the residual unhydrated cementitious materials particles, such as: cement, activated slag [7], FA [14] and MK [7,15] can further react with water, which mainly comes from the vaporization of moisture [2] and dehydration of calcium hydroxide (CH), leading to the formation of dense hydration products [16,17,18].

Fly ash (FA), as the most typical SCM, has been extensively studied for the physical and chemical modification of cement hydration, as well as its hardened properties [10,19,20,21], and numerous reports have confirmed that the introduction of FA can increase the high-temperature resistance of concrete. However, the studies of the effects of FA and its combination with other SCMs, on the properties of HPC subjected to high temperature is still very limited. Metakaolin (Al_2_O_3_·2SiO_2_·2H_2_O), as a kind of high activity SCMs, can form anhydrous aluminum silicate (Al_2_O_3_·2SiO_2_) by dehydration at a temperature range of 600 to 900 °C [7,19,22]. This compound makes it possible to improve the high-temperature resistance of HPC containing SCMs subjected to high temperatures. The increasing application of various types of SCMs in concrete makes the high-temperature resistance of HPC even more complex [2,4]. Therefore, the studies of high-temperature resistance of concrete containing SCMs become an urgent issue.

We have previously investigated the effects of recipes of SCMs on the workability (slump), mechanical (residual strength) and physical properties (loss mass, water absorption, and porosity) of concrete after exposure to high temperature [23]. Residual strength of a concrete with 20% FA-10% UFFA-5% MK at 800 °C maintained 71.3% of its original strength, and residual strength of MK-UFFA and MK-UFFA-FA concrete at 1000 °C retained 18.7 MPa and 23.3 MPa, respectively [23]. However, the effects of recipes of SCMs on the thermal performance and microstructure evolution of concrete subjected to high temperatures are still unclear.

In this study, the effects of SCMs combinations on the thermal performance, microstructure, the crystalline and amorphous phases evolution of concrete subjected to high temperatures are investigated. Four types of HPC formulations were prepared: HPC modified with FA, HPC modified with UFFA, HPC modified with UFFA-MK, and HPC modified with FA-UFFA-MK. The experimental results explain the micro-mechanisms of concrete containing SCMs at high temperatures, and provide theoretical support for the design of concrete with high temperature resistance requirement.

## 2. Materials and Methods

### 2.1. Materials

Portland cement (P·O 52.5R) was used in this study. Table 1 lists the properties of cement according to the Chinese Standard GB 175-2007 [24], three types of SCMs used to improve the high-temperature resistance of concrete (Figure 1), were from a local manufacturer. Figure 2 presents the chemical composition of the SCMs, which complied with the requirements of the Chinese Standard GB/T 51003-2014 [25].

Broken granite gravel with continuous grading was used. The crushed index, maximum particle size and apparent density of the coarse aggregate are 8.7%, 10 mm and 2.895 g/cm^3^, respectively. The fine aggregate used was natural river sand with a fineness modulus of 2.6. Table 2 presents the grain size distribution of aggregates. The properties of the fine and corsage aggregate complied with the Chinese standards GB/T 14684-2011 [26] and GB/T14685-2011 [27], respectively.

Polycarboxylate-based superplasticizer (SP) was used, with a solid content and water-reducing ratio of 35% and 28%, respectively.

### 2.2. Mixing Process

In order to investigate the effects of SCMs combinations on the thermal performance, microstructure, the crystalline and amorphous phases evolution of concrete subjected to high temperatures, five different mix proportions were considered in this study. Table 3 presents the mix proportions of concrete. In the mixing process, all parameters were kept constant apart from the content of SCMs. The workability of concrete is fundamentally controlled by the recipes of SCMs. In order to further understand the effects of recipes of SCMs on the workability of concrete, please refer to our published previously [23].

### 2.3. Testing Method

For residual compressive strength test, a specimen with dimensions of 100 × 100 × 100 mm was dried to constant weight after 28 d of curing, and then heated in an electrically controlled furnace (Luoyang Zhipu Furnace Co., Ltd., Luoyang, China), as shown in Figure 3. When they reached the set temperatures (400, 600, 800 or 1000 °C), samples were kept at constant temperature for 3 h, and then naturally cooled to room temperature (RT) before testing.

The thermal conductivity of concrete after exposure to high temperatures was tested by thermal constant analyzer (Hot Disk, TPS 2500S, Gothenburg, Sweden) [8,17]. For thermal conductivity test, a cube sample was cut into two pieces (50 × 50 × 70 mm), and a test metal probe was sandwiched between the two pieces of specimens. Two samples (50 × 50 × 70 mm) were placed face to face during the test, similar to a sandwich structure. Figure 4 presents the schematic diagram of a thermal conductivity test. In order to ensure the accuracy of the test, the metal probe with the largest diameter should be selected as long as the probe meets the size of the test sample. A cylindrical metal probe with dimension of Ø10 mm × 15 mm was used in this study to measure the thermal conductivity of concrete. The input power and test time of thermal conductivity test were 50 m W and 160 s, respectively [5]. Kept the temperature at (20 ± 3 °C) and relative humidity at (20–40 %) during the test.

The microstructure of concrete subjected to high temperatures was characterized by Quanta 600 FEG scanning electron microscope (SEM, FEI, Hillsboro, Oregon, USA). Specimens were broken into small pieces and surfaces were coated by gold to increase its electrical conductivity before testing. X-ray diffraction (XRD) was used to identify the evolution of the crystalline phases of concrete subjected to high temperatures. Nickle-filtered Cu-Ka radiation at 40 kV and 20 mA were used throughout in a PW1390 diffractometer (PANalytical B.V., Almelo, The Netherlands). Scanning speed of 0.020 °/s was used. Differential Thermal Analysis (DTA) thermal analysis (DTA1/1600, Mettler-Toledo, Zurich, Switzerland) was used to explain the amorphous phases in the hydration products of the concrete at ambient and at high temperatures. Samples were analyzed under an oxygen atmosphere (100 ml/min) at a heating rate of 10 °C /min using alumina crucibles.

## 3. Results and Discussion

### 3.1. Thermal Conductivity

The thermal conductivity of concrete is an essential parameter for delaying the temperature increase when subjected to high temperatures [5,8,17]. Figure 5 exhibits the thermal conductivity of concrete at RT and high temperatures.

From Figure 5, it can be seen that the thermal conductivity of concrete slightly increased at 400 °C and significantly dropped at 800 °C. In fact, residual strength of concretes increased at 400 °C [23], strongly supporting that some hydration products were also formed, which can fill the pores and even more than compensate for the loss of strength caused by the decomposition of some hydration products [8,21]. The thermal conductivity of concrete is mainly related to the porosity and pore size [8], a significantly drop of thermal conductivity at 800 °C due to the decomposition of CH and desiccation of pore system [28], it will be discussed by the the XRD analysis and microstructure observations of concrete in Section 3.2 and Section 3.4. A significantly drop of thermal conductivity induced micro-cracking and increased the porosity of concrete, it was also supported by the porosity test results of concrete in our previous paper [23].

It can be observed that the thermal conductivity of the concretes approximately increased by 1%–5% than that of the concrete at RT. Compared with the thermal conductivity of concretes at RT, thermal conductivity of concretes at 400 °C increased by 2.1% (reference concrete), 3.4% (HPC-1), 1.8%(HPC-2), 3.8%(HPC-3), 4.8% (HPC-4). We also found that the thermal conductivity of concretes modified with SCMs was higher than that of the reference concrete. At 400 °C, with the addition of 20% FA (wt %), the thermal conductivity of the sample (HPC-1) slightly increased to 1.5 W/(m**·**K). Replacing FA with UFFA can further increase the thermal conductivity (HPC-2) to 1.7 W/(m**·**K), which is mainly due to the UFFA’s lower fineness, which can make the HPC denser. For the samples containing 30% UFFA-5% MK (HPC-3) and 20% FA-10% UFFA-5% MK (HPC-4), thermal conductivity increased to 1.9 W/(m**·**K) and 2.2 W/(m**·**K) at 400 °C, respectively. It can be concluded that the difference of thermal conductivity is much more evident between concretes than the difference attributed to the temperature itself when the temperature below 400 °C. At 800 °C, thermal conductivity of the concretes modified with UFFA-MK and FA-UFFA-MK fell to 0.3 W/(m**·**K) and 0.4 W/(m**·**K), respectively. This may be due to the internal structure of concretes had destructed at 800 °C.

### 3.2. XRD Analysis

Figure 6 shows the XRD patterns of the reference concrete and HPC-4 subjected to high temperatures. Typical reflections associated with larnite (C_2_S), portlandite (CH), ettringite (AFt), calcite (CaCO_3_), brown millerite (Ca_4_Al_2_Fe_2_O_10_) and lime can be found in the reference specimen, as shown in Figure 6a. Some reflections (AFt) disappeared after heating, this could be attributed to the decomposition of AFt after heating to 70 °C, similar results have been obtained by Molay et al. [2] and Rashad et al. [29]. Up to 400 °C, a progressive reduction of the intensity of the peak of portlandite was obtained, which disappears at 800 °C. Previous research work by Alonso et al. [6] obtained a similar observation that the CH crystal began to decompose between 350 °C and 600 °C. The brownmillerite was found in all samples and the same for larnite. At 1000 °C, reflection peak related to calcite practically disappeared, while lime was also well detected in the sample, which could be attributed to the transformation of portlandite and calcite.

Compared with the reference concrete, the reflections (such as mullite, gismondine and calcium aluminum oxide, etc.) can be found in the concrete containing FA-UFFA-MK (HPC-4) after heating, as shown in Figure 6b. This is a good explanation for the highest residual strength obtained of HPC-4 subjected to high temperatures, as illustrated in [23].

Figure 6c,d presents the XRD patterns of specimens (reference concrete, HPC-2 and HPC-4) at RT and 400 °C, respectively. Calcite could be found in the specimens and even increased in intensity up to 400 °C. Additionally, reflection peaks of calcium aluminum oxide appeared in HPC-2 and HPC-4 mixtures after heating. This could be attributed to the water from the decomposition of hydration products let to rehydrate the anhydrous cement particles [2], which increased the residual strength of concrete at 400 °C. Similar results have been reported by Wang et al. [8].

### 3.3. DTA Analysis

Figure 7 shows the variations of the DTA thermograms of reference concrete, HPC-2 and HPC-4 at RT and high temperatures. For the reference concrete, the first peak (# 1) was located at about 80 °C and 110 °C, which is attributed to the loss of free water and to the decomposition of the nearly amorphous calcium silicate hydrates. The second peak (# 2) located at 400–450 °C due to the decomposition of CH [22]. The third peak (# 3) observed at 700–800 °C represents the decarbonation of the limestone addition present in the cement used and the loss of the remaining water from the decomposition of the hydrated calcium silicate.

For the HPC containing fly ash and MK, the peak (# 4) between 160 °C and 180 °C is attributed to the dehydration of the C-A-H and the hydrated calcium silicoaluminates (C-A-S-H) [19]. The peak (# 5) represents the oxidation of Fe_2_O_3_ oxide and the decomposition of unburnt coal residues [22]. Additionally, it can be seen that the HPC-2 and HPC-4 show a higher peak (# 2) due to the addition of UFFA and FA-UFFA-MK that generates a lot of CH [22].

### 3.4. Microstructure Observations

Figure 8 shows the microstructure of specimen (the reference concrete, HPC-2 and HPC-4) subjected to RT and 400 °C, respectively. Higher internal structure and more hydration products of specimens could be found on specimens (HPC-2 and HPC-4) than on the reference concrete, this due to the pozzolanic reaction of the SCMs, as well as combined usage of SCMs that optimized particle grading [14]. Therefore, HPC modified with SCMs showed higher residual strength than that of the control mix, as illustrated in [23].

Needle-like crystals of ettringite, CH and C-S-H gel can be found in Figure 8a. While the concretes after exposure to 400 °C, it can be seen that the CH crystal began to decompose, as seen in Figure 8b. In addition, the microstructure of concrete at 400 °C showed more compactness in comparison to the concrete at RT, which could be attributed to the water from the decomposition of some hydration products at 400 °C, resulting in rehydrate the anhydrous cement particles existed in the capillary pores [22]. This was also supported by the thermal conductivity of concrete, as discussed in Section 3.1.

From Figure 8c–f, it can be observed that the hydration products (CH and C-S-H gel) of cementitious materials began to decompose, and even increased in density as can be seen in the sample exposed to 400 °C. As UFFA was added as SCM (HPC-2), the microstructure of concrete presented more compactness, since UFFA is quite chemically stable below 500 °C [30], the benefits of using UFFA as SCM, should be resulted from the fineness and pozzolanic reaction of the composites binding materials, which promote the effects of rehydration of the anhydrous cement on the heat resistance of concrete. Interestingly, when MK was added as SCM together with FA and UFFA (HPC-4) showed denser internal structure, this could be attributed to the higher amount of gel-like hydration products and lower CH crystal contents [1,30], which increased high-temperature resistance of concrete. Indeed, it has been reported that the pozzolanic reaction of amorphous aluminosilicate present in MK with CH generated by cement hydration can form an additional amount of C-S-H that has low Ca/Si with high strength [22,30], as well as calcium aluminates hydrate (C-A-H) phases that deposit in the pore system [22]. Both of which will bridge the pore system and lead to a reduction of the thermal stresses generated around the pores [22]. In addition, the fineness of the UFFA further reduce the thickness of C-S-H gel and improve the heat resistance of concrete. Furthermore, calcite could be found in the specimens and even increased in intensity at 400 °C [31,32], as illustrated in Section 3.2. Leading to a slightly increased in residual strength of the specimen subjected to 400 °C.

## 4. Conclusions

This paper presents a study of the effects of SCMs combinations on the thermal performance, microstructure, the crystalline and amorphous phases evolution of concrete subjected to high temperatures. Based on the experimental tests and the microstructure analyses, the following conclusions can be drawn:

1Combined usage of SCMs improved the high temperature resistance of concrete after subjected to high temperatures. Formation of the secondary gel and filling capacity by combined usage of SCMs, making the microstructure denser, can exclude propagation of cracks at high temperature.2The thermal conductivity of both the reference concrete and HPC modified with SCMs increased slightly at 400 °C and reduced significantly at 800 °C. Thermal conductivity of concretes modified with SCMs was higher than that of the reference concrete. The thermal conductivity of the HPC modified with MK-UFFA and MK-UFFA-FA at 800 °C dropped to 0.3 W/(m·K) and 0.4 W/(m·K), respectively.3The residual strength of concrete at 400 °C showed a slight increase due to the water result in the decomposition of some hydration products let to rehydrate the anhydrous cement particles. The internal structure cracked and damaged owing to the decomposition of C-S-H gel and CH at 600 °C. For the composite 20% FA-10% UFFA-5% MK mixture after heating to 1000 °C, the unreacted MK particles in the microstructure transformed to mullite as identified in the XRD patterns.4The combined usage of SCMs to concrete contributes to the enhancement of high temperature resistance due to the generation of C-S-H gel and densification of the microstructure. FA not only could be prominently effective at concrete after exposure to high temperatures but works progressively alongside other pozzolanic materials, such as MK. Filling empty pores with different sizes and promoting pozzolanic reactions are two of the reasons that improved the high temperature resistance of concretes.

## Figures and Tables

**Figure 1 materials-13-01833-f001:**
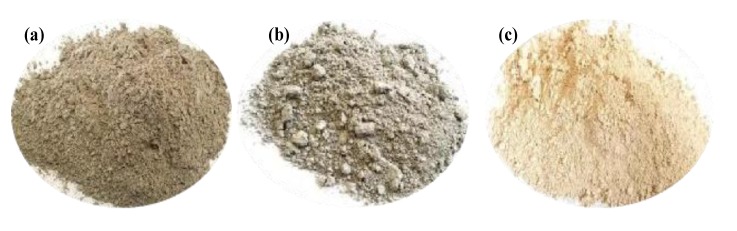
Supplementary cementitious materials (SCMs) used in concrete mixes: (**a**) fly ash (FA), (**b**) ultr-fine fly ash (UFFA) and (**c**) metakaolin (MK).

**Figure 2 materials-13-01833-f002:**
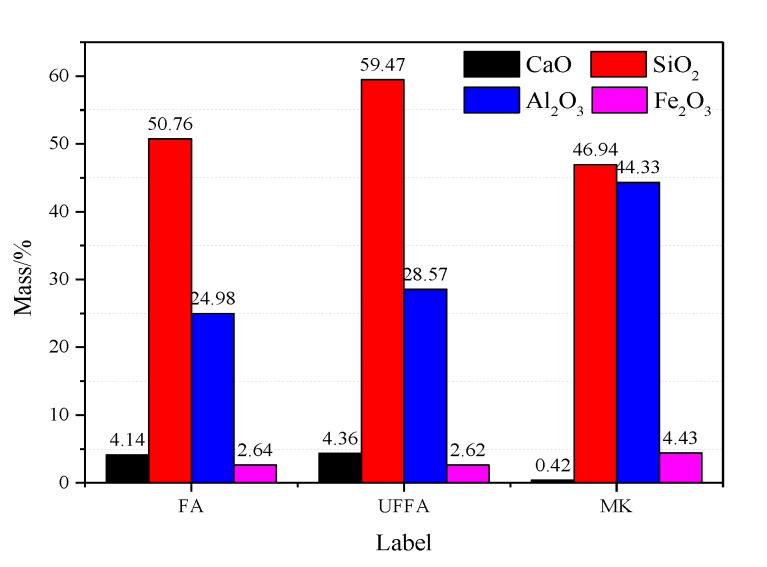
Chemical composition of FA, UFFA, and MK (mass %).

**Figure 3 materials-13-01833-f003:**
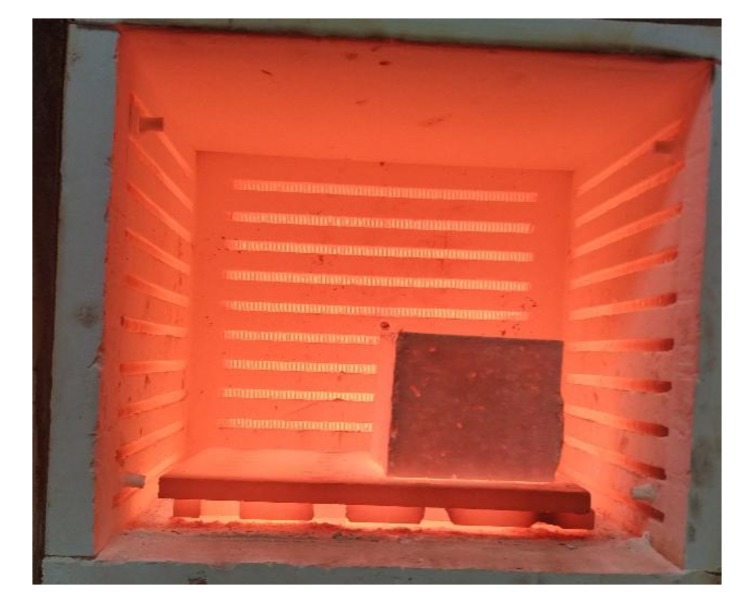
Concrete in a muffle furnace.

**Figure 4 materials-13-01833-f004:**
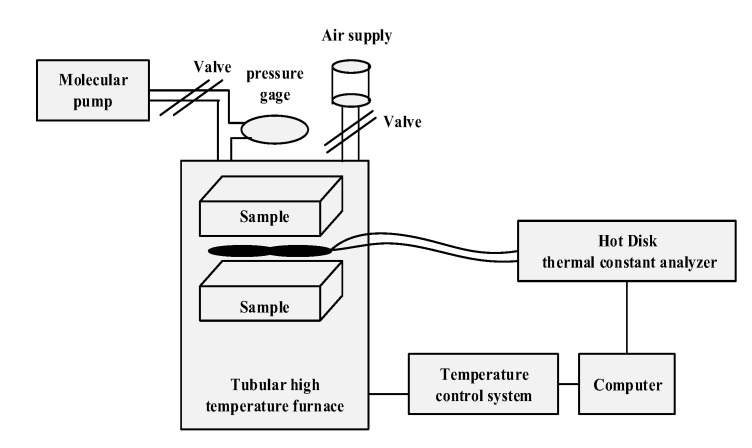
The schematic diagram of thermal conductivity measurement.

**Figure 5 materials-13-01833-f005:**
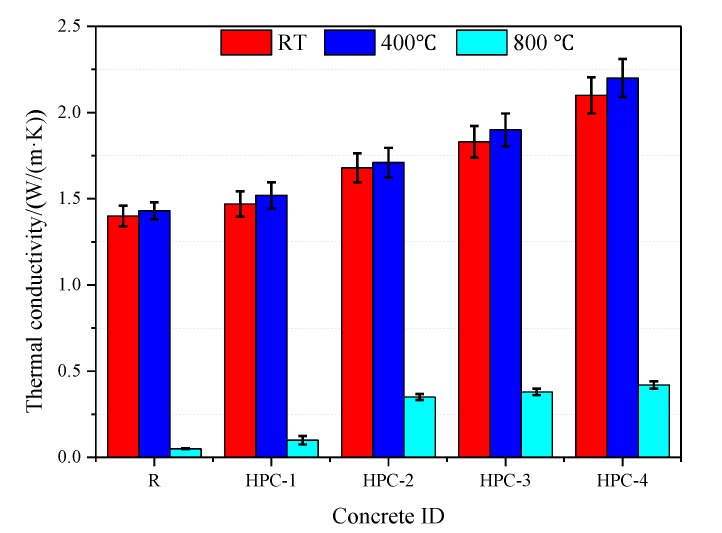
Thermal conductivity of concrete subjected to high temperatures.

**Figure 6 materials-13-01833-f006:**
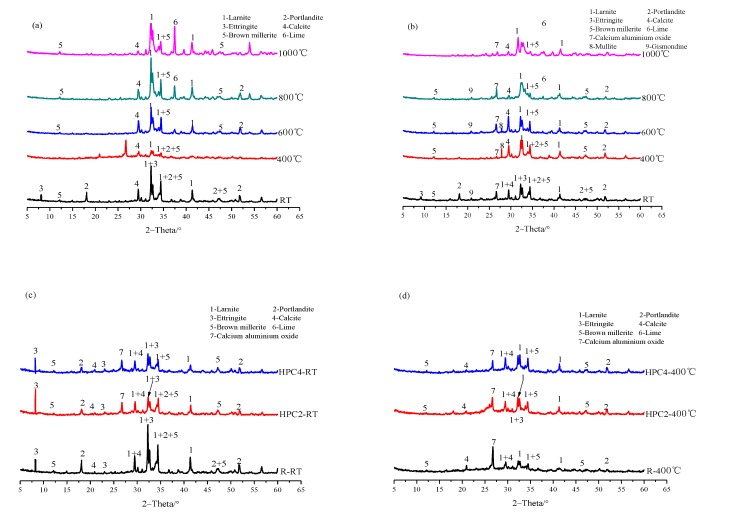
XRD patterns of concrete: (**a**) and (**b**) refer to the reference concrete and HPC-4 subjected to various temperatures, respectively; (**c**) and (**d**) refer to the specimens (the reference concrete, HPC-2 and HPC-4) under RT and 400 °C, respectively.

**Figure 7 materials-13-01833-f007:**
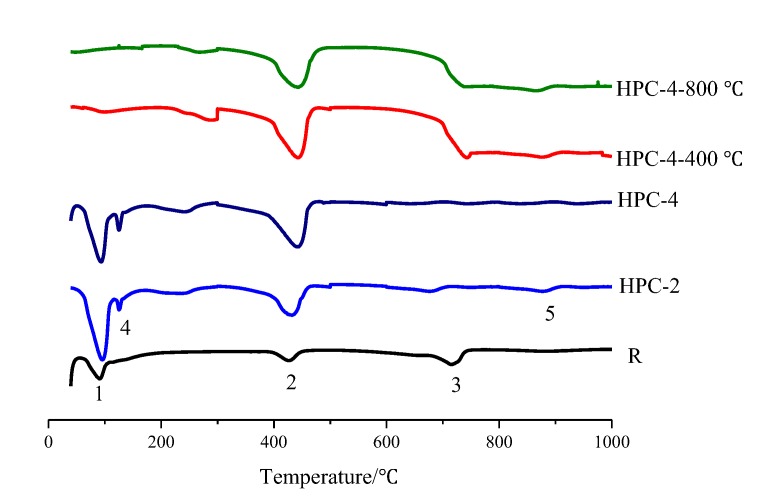
Differential Thermal Analysis (DTA) curves for the reference concrete, HPC-2 and HPC-4 after 28 d of curing.

**Figure 8 materials-13-01833-f008:**
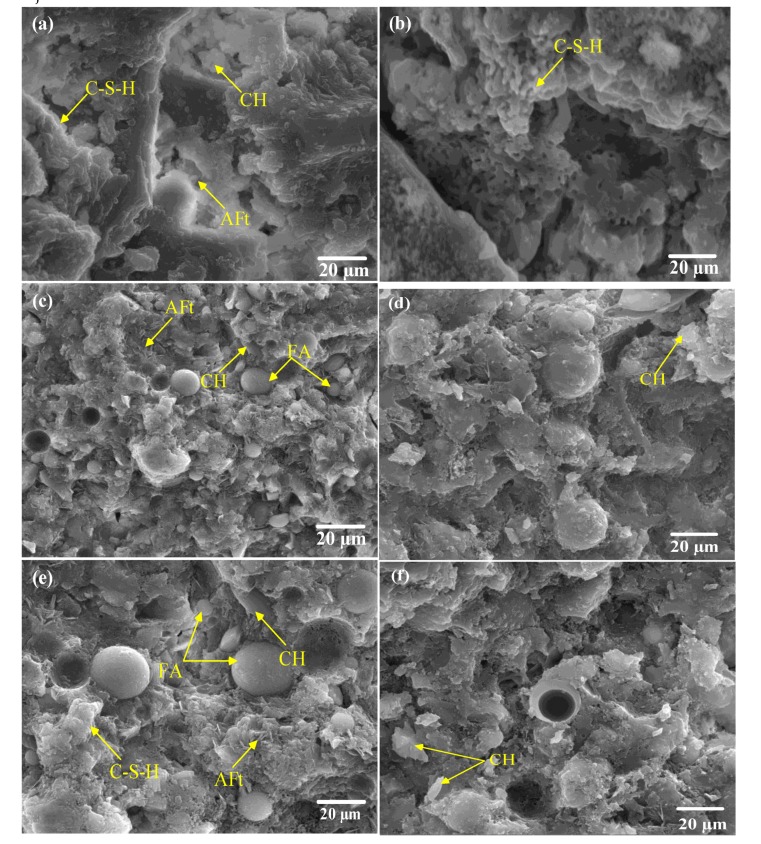
The microstructure alteration of concrete after heating: (**a**) and (**b**) refer to the reference concrete subjected to room temperature (RT) and 400 °C, respectively; (**c**) and (**d**) refer to HPC-2 subjected to RT and 400 °C, respectively; (**e**) and (**f**) refer to HPC-4 subjected to RT and 400 °C, respectively.

**Table 1 materials-13-01833-t001:** Properties of cement.

Density (kg/cm^3^)	Specific surface properties (m^2^/kg)	Setting time (min)		Compressive strength (MPa)		Flexural strength (MPa)	
		Initial setting	Final setting	3 d	28 d	3 d	28 d
3.06	461.2	115	150	28.6	60.1	5.6	9.1

**Table 2 materials-13-01833-t002:** The grain size distribution of aggregates.

Sieve size (mm)		16	9.5	4.75	2.36	1.18	0.6	0.3	0.15
Cumulative percentage retained, by mass (％)	Coarse	0	12.5	87.8	98.2	-	-	-	-
	Fine	-	-	3.1	15.7	22.1	43.2	88.4	96.8

**Table 3 materials-13-01833-t003:** Mix proportions of concrete (kg/m^3^).

Concrete ID	W/B	Cement	FA	UFFA	MK	Sand	Gravel	Water	SP
R	0.21	600	-	-	-	634	1126	126	6
HPC-1	0.21	480	120	-	-	634	1126	126	6
HPC-2	0.21	420	-	180	-	634	1126	126	6
HPC-3	0.21	390	-	180	30	634	1126	126	6
HPC-4	0.21	390	60	120	30	634	1126	126	6

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
