# Peer review of "Effects of Combined Usage of Supplementary Cementitious Materials on the Thermal Properties and Microstructure of High-Performance Concrete at High Temperatures"

_materials, 2020, doi:10.3390/ma13081833_

Round 1
Reviewer 1 Report
Dear Authors,
Most of my comments are on the attached PDF document. In general, I consider that the paper is interesting, related to the topic of the journal, and worth to be published after mayor revisions.
English should be revised by a native, as there are many sentences that are difficult to understand. Apart from that, some tests must be better explained, for instance, thermal conductivity test. In fact, an statistical analysis should be included in this part, showing, at least, the 95% confidence intervals with error bars on fig. 5.
Furthermore, I felt that the paper is too much based on a previous paper, denoting that, maybe, the information included in this paper should have already been included on that one. Thus, I recommend The Authors to repeat the compressive strength tests, as it would enhance the actual paper, strengthening also the previous one.
Finally, I am not an expert on the microstructure of the concrete at different temperatures, but XRD, DTA and SEM analyses seem to be well addressed and studied.

Reviewer 2 Report
The paper is interesting and sound this reviewer just has a two comments:
In order to increase the value of the paper the first paragraph of the introduction must mention several cases of concrete structures that were submitted to high temperatures.
The testing method used different temperatures (400, 600, 800 and 1000 ℃) during 3 hours, and then naturally cooled. Why this method ? Usually in a fire its very unusual that a concrete structure will be 3 hours at high temperature before firefighters arrive. Also when they arrive they use water that will make high temperature concrete to suffer a thermal shock. Why not also simulate this shock ?
Round 2
Reviewer 1 Report
Thank you very much for taking into account my recommendations.